# Effects of a Home-Based Lifestyle Intervention Program on Cardiometabolic Health in Breast Cancer Survivors during the COVID-19 Lockdown

**DOI:** 10.3390/jcm10122678

**Published:** 2021-06-17

**Authors:** Valentina Natalucci, Carlo Ferri Marini, Marco Flori, Francesca Pietropaolo, Francesco Lucertini, Giosuè Annibalini, Luciana Vallorani, Davide Sisti, Roberta Saltarelli, Anna Villarini, Silvia Monaldi, Simone Barocci, Vincenzo Catalano, Marco Bruno Luigi Rocchi, Piero Benelli, Vilberto Stocchi, Elena Barbieri, Rita Emili

**Affiliations:** 1Department of Biomolecular Sciences, University of Urbino Carlo Bo, 61029 Urbino, Italy; valentina.natalucci@uniurb.it (V.N.); carlo.ferrimarini@uniurb.it (C.F.M.); francesco.lucertini@uniurb.it (F.L.); giosue.annibalini@uniurb.it (G.A.); luciana.vallorani@uniurb.it (L.V.); davide.sisti@uniurb.it (D.S.); roberta.saltarelli@uniurb.it (R.S.); marco.rocchi@uniurb.it (M.B.L.R.); piero.benelli@uniurb.it (P.B.); 2U.O.C. Cardiologia/UTIC, Ospedale Santa Maria della Misericordia, Area Vasta n.1, 61029 Urbino, Italy; marco.flori@sanita.marche.it (M.F.); francesca.pietropaolo@sanita.marche.it (F.P.); 3Epidemiology Unit, Department of Research, Fondazione IRCCS Istituto Nazionale dei Tumori, 20133 Milano, Italy; a.villarini@istitutotumori.mi.it; 4U.O.C. Oncologia Medica, ASUR Area Vasta 1, Ospedale Santa Maria della Misericordia di Urbino, 61029 Urbino, Italy; silvia.monaldi@sanita.marche.it (S.M.); vincenzo.catalano@sanita.marche.it (V.C.); rita.emili@sanita.marche.it (R.E.); 5U.O.C. Patologia Clinica, Ospedale Santa Maria della Misericordia, Area Vasta n. 1, 61029 Urbino, Italy; simone.barocci@sanita.marche.it; 6Department of Human Sciences for the Promotion of Quality of Life, University San Raffaele, 20132 Roma, Italy; vilberto.stocchi@uniroma5.it

**Keywords:** COVID-19, breast cancer survivors, home-based lifestyle intervention, Mediterranean diet, exercise, cardiotoxicity, cardiovascular fitness, heart rate variability, secondary and tertiary prevention

## Abstract

This study aimed to evaluate the cardiometabolic effects of a home-based lifestyle intervention (LI) in breast cancer survivors (BCSs) during the COVID-19 lockdown. In total, 30 BCSs (women; stages 0–II; non-metastatic; aged 53.5 ± 7.6 years; non-physically active; normal left ventricular systolic function) with a risk factor for recurrence underwent a 3-month LI based on nutrition and exercise. Anthropometrics, Mediterranean diet adherence, physical activity level (PAL), cardiorespiratory fitness (*V*O_2max_), echocardiographic parameters, heart rate variability (average standard deviation of NN intervals (ASDNN/5 min) and 24 h very- (24 hVLF) and low-frequency (24 hLF)), and metabolic, endocrine, and inflammatory serum biomarkers (glycemia, insulin resistance, progesterone, testosterone, and high-sensitivity C-reactive protein (hs-CRP)) were evaluated before (T0) and after (T1) the LI. After the LI, there were improvements in: body mass index (kg/m^2^: T0 = 26.0 ± 5.0, T1 = 25.5 ± 4.7; *p* = 0.035); diet (Mediet score: T0 = 6.9 ± 2.3, T1 = 8.8 ± 2.2; *p* < 0.001); PAL (MET-min/week: T0 = 647 ± 547, T1 = 1043 ± 564; *p* < 0.001); *V*O_2max_ (mL·min^−1^·kg^−1^: T0 = 30.5 ± 5.8, T1 = 33.4 ± 6.8; *p* < 0.001); signs of diastolic dysfunction (participants: T0 = 15, T1 = 10; *p* = 0.007); AS-DNN/5 min (ms: T0 = 50.6 ± 14.4, T1 = 55.3 ± 16.7; *p* = 0.032); 24 hLF (ms^2^: T0 = 589 ± 391, T1 = 732 ± 542; *p* = 0.014); glycemia (mg/dL: T0 = 100.8 ± 11.4, T1 = 91.7 ± 11.0; *p* < 0.001); insulin resistance (HOMA-IR score: T0 = 2.07 ± 1.54, T1 = 1.53 ± 1.11; *p* = 0.005); testosterone (ng/mL: T0 = 0.34 ± 0.27, T1 = 0.24 ± 0.20; *p* = 0.003); hs-CRP (mg/L: T0 = 2.18 ± 2.14, T1 = 1.75 ± 1.74; *p* = 0.027). The other parameters did not change. Despite the home-confinement, LI based on exercise and nutrition improved cardiometabolic health in BCSs.

## 1. Introduction

The COVID-19 pandemic is a global public health emergency. On 20 February 2020, the first patient diagnosed with COVID-19 in Italy developed respiratory failure and was admitted to the intensive care units of Lombardy, a region of northern Italy. Since then, the number of cases recorded increased steadily throughout the country, leading Italy to be the second most affected country in the world as of 27 March 2020 [1]. The first wave of the COVID-19 pandemic was a particularly worrying time for vulnerable groups with pre-existing health conditions, such as breast cancer (BC) survivors (BCSs), because the mandatory directives for the first lockdown (in Italy between 9th March to 3rd May 2020) certainly altered the daily routine in clinical settings [2]. BC is the most common neoplasm worldwide, representing the primary cause of death due to neoplasms [3]. In Italy, it represents 14.6% of all new tumor diagnoses. The trend of incidence rate in Italy slightly increased (+0.3% per year), while mortality has progressively decreased (−0.8% per year) probably due to significant improvements in screening protocols, diagnosis, and treatment over the past few decades [4]. The 5-year survival rate in Italy is 87% [4,5]. These numbers, which are constantly growing, require careful evaluation of the health and social impact of the cancer follow-up and lifestyle intervention in terms of improving the prognosis and reducing the risk of relapse [6]. In this light, during the pandemic emergency, many of the factors that impact risk of recurrence, such as lifestyle choices, have been modified. In this context, particular attention should be focused on patients at high risk of BC recurrence that experienced metabolic syndrome. Indeed, metabolic syndrome is associated with a 17% increase in BC risk [7], three-fold increase in BC recurrence, and an about two-fold increase in BC specific mortality [8]. Moreover, metabolic syndrome has been associated with chronic low-grade systemic inflammation with an interplaying role in the BC initiation, progression, poor prognosis and recurrence [9]. BC is considered as a heterogeneous disease. Indeed, molecular characteristics include activation of human epidermal growth factor receptor 2 (HER2, encoded by ERBB2), activation of hormone receptors and/or BRCA mutations. Thus, the potential treatment strategies can differ according to BC molecular subtype. The BC management is based on a multidisciplinary approach and includes local (surgery and radiation therapy) and circulating and oral therapy (e.g., bone stabilizing agents, poly (ADP-ribose) polymerase inhibitors and endocrine-, chemo-, anti-HER2-, immuno-therapy) [10]. In addition to further therapeutic advances for BCSs that include the above-mentioned medications, exercise, counselling, physical and occupational therapy, and alternative and complementary therapies should also be taken into consideration and remain the global challenge in BCS care for the future [11].

Among lifestyle behaviors, two interventions have been considered as strong allies to cancer patients and survivors, as they positively impact body weight, physical fitness, fatigue, depressive symptoms, anxiety, inflammatory profile and quality of life [12,13,14,15,16,17,18,19]: nutrition and physical activity (PA). In this regard, unhealthy eating habits and a sedentary lifestyle are the main causes of metabolic syndrome and may worsen the inflammation conditions and symptoms in BC patients who have received chemotherapy and survivors [20,21,22,23,24].

Diet, considered as the sum of all food that is consumed, is known to influence BC risk and mortality [25]. In the most recent report of the World Cancer Research Fund International (WCRF)/American Institute for Cancer Research (AICR), it is estimated that, encompassing the 13 most common cancers, 29% of cases could have been prevented by a healthy lifestyle. Actually, WCRF/AIRC 2018 focused on a diet rich in wholegrains, vegetables, fruit and beans, limited consumption of “fast food” and other processed foods high in fat, starches or sugars, red and processed meat, sugar sweetened drinks, alcohol and the non-use of supplements [26].

Regular PA, i.e., exercise, may contribute to reduce the risk of comorbid conditions such as cardiovascular disease (CVD), which is a growing concern among BCSs [27,28,29,30]. Current exercise prescription guidelines in people who have been treated for cancer include avoiding physical inactivity and performing at least 30 min of moderate-intensity aerobic exercise at least three times per week to obtain health benefits. Moreover, the exercise program should also include resistance exercises performed at least two times per week [27,31,32]. Exercise provides great benefits to patients when delivered in supervised group- and clinic-based settings. However, in the age of COVID-19, exercising in gym facilities had the risks for virus transmission through aerosol and surface contact among the exercisers. A recent study showed that BCSs increased sedentary behaviors and decreased PA levels (PALs) during the pandemic lockdown [33,34,35]. In these circumstances, it could be useful to adapt the usual methods of exercise delivery to home-based exercise programs [27] to counteract detrimental effects of home-confinement [36].

The pandemic lockdown with home-confinement probably exacerbated health risks, since it might have also changed food consumption patterns and PALs, potentially impacting on the longer-term health of BC populations. Indeed, it has been suggested that lifestyle interventions are needed to ameliorate BC outcomes and to reduce the risk of comorbidities and recurrences [37,38].

The aim of this study was to describe the effects of a home-based lifestyle intervention in women with BC at high risk of recurrence due to metabolic or endocrine disorders with detrimental effects on cardiometabolic health (cardiac function, heart rate variability (HRV), cardiorespiratory fitness, metabolic prognostic parameters, and anti-inflammatory response) during the COVID-19 lockdown.

## 2. Materials and Methods

### 2.1. Study Population

Data from thirty BCS women participating in the MoviS trial (approval of the local Ethics Committee, permission number: 21/19 10 July 2019, ClinicalTrials.gov reference number: NCT04818359) were analyzed. Briefly, participant inclusion criteria were: ≤12-month post-surgery and post chemo- or radio-therapy adjuvant; stage 0 to III BC without metastases or recurrences diagnosis at recruitment in follow-up; aged 30–70 years; non-physically active (i.e., not engaged in at least 60 min/week of structured exercise during the previous 6 months); with a risk factor for recurrence. As reported in previous studies [39,40,41,42], the risk factors for recurrence were: body mass index (BMI) at diagnosis ≥25 kg/m^2^, testosterone ≥0.4 ng/mL; serum insulin ≥25 µU/mL (170 pmol/L); metabolic syndrome (at least 3 of the following 5 factors): a, glycemia ≥100 mg/dL (6.05 mmol/L); b, triglycerides ≥150 mg/dL (1.69 mmol L); c, HDL-C <50 mg/dL (1.29 mmol/L); d, waist circumference ≥80; e, blood pressure ≥130/85 mmHg. Exclusion criteria were: disabling pneumological, cardiological, neurological, orthopedic comorbidities, or mental illnesses that prevent the exercise performance; treatment with beta blockers, non-dihydropyridine calcium channel blockers, or amiodarone due to their potential effect on heart rate response to exercise; treatment with antidepressant drugs.

Recruitment occurred in January 2020 from the Santa Maria della Misericordia Hospital of Urbino (PU) in the Marche region (central Italy). The purpose of the protocol was explained to the patients and written informed consent was obtained from each patient before inclusion.

### 2.2. Study Design

The MoviS trial is an open randomized trial with two parallel groups. Enrolled participants, who originally were randomly allocated to an intervention arm (IA) and a control arm (CA), received a 3-month lifestyle intervention and several assessments were performed before (T0) and after (T1) the intervention. Due to the imposed COVID-19 pandemic restrictions, after the approval of the institutional ethics committee, the study protocol was amended (Protocol N. 29/20 22.04.2020) (Figure 1). The forced changes in the study protocol made the difference between IA and CA interventions negligible (please, see “Lifestyle intervention” for details), providing similar adaptations between groups (Appendix A). Therefore, in the present article, due to the lack of meaningful differences between the two interventions, the results of the two groups were combined.

### 2.3. Lifestyle Intervention

At the study enrollment, BCSs (IA and CA) received 3-month lifestyle (nutrition and exercise) educational counselling: in the 15 days prior to the start of the intervention phase, motivational interviews were organized, structured in meetings lasting about one hour (45 min group and 15 personalized minutes) with patients for a common phase of psychological support by the psycho-oncologist of the U.O.C. of Oncology [43]. At the end of this phase, recommendations on nutritional pathways and PA were indicated by an oncology nutritionist and an exercise oncology expert. They were given a brochure with the oncological lifestyle recommendations based on the WCRF 2018 and the recent guidelines on nutritional and exercise for BC patients, approved by the Ministry of Health 2017 and 2019 [26,31,32,44,45].

The motivational interviews were performed before the intervention and during the entire experimentation period oncology nutrition and exercise experts stimulated the patients weekly by a social chat to adhere to the change in lifestyle with a short message service at the beginning of each session week.

The nutritional advice was based on the Mediterranean diet, which consisted of 3 meals and 2 snacks per day. The meals had to mainly contain whole grains, legumes, vegetables and seasonal fruit; fish was the most frequent animal protein and other animal proteins were included in a moderate way in the diet. In particular, the consumption of preserved meats, sugary drinks, foods with refined ingredients and ultra-processed foods were discouraged. The participants also participated in cooking sessions, in which they were invited to cook and eat some dishes prepared according to the WCRF/AICR recommendations and inspired by a Mediterranean diet available on the DianaWeb platform [46,47].

In addition to the counselling, the IA was supposed to receive a 3-month remotely (1 session per week) and on-site (2 sessions per week) supervised aerobic exercise training program having progressive increases in exercise intensity (from 40% to 70% of heart rate reserve) and duration (from 20 to 60 min). Exercise intensity and duration were gradually increased to reach and exceed the recommended quality (exercise intensity) and quantity (volume) of aerobic exercise for BCSs [31,32]. However, due to the imposed COVID-19 pandemic restrictions, from the 4th week the type of supervision was adapted to solely remotely supervised exercise (3 sessions per week). The supervision was performed weekly, using phone calls from the exercise specialist, who provided the weekly exercise prescription and personalized feedback according to the training logs.

Both remotely and on-site supervised training sessions consisted of aerobic exercise (i.e., walking, running, or cycling). On-site supervised sessions were performed in a gym using a treadmill or stationary bikes, whereas the remotely supervised sessions were performed both indoors and outdoors according to participants’ possibilities and preferences. Regardless of the exercise modality, the sessions were performed at individualized exercise intensities (e.g., walking speed and grade or cycling wattage), allowing each participant to reach and maintain the prescribed target heart rate (HR) during the training sessions.

### 2.4. Assessments

#### 2.4.1. Anthropometrics and Body Composition

Weight was measured to the nearest 0.1 kg on an electronic scale with the patient wearing a hospital gown and no shoes. Height was measured to the nearest 0.5 cm with a fixed stadiometer. Waist circumference was measured at the midpoint between the lower margin of the last palpable rib and the iliac crest. Bioimpedance measurements were recorded by DC430MA DC 430 (Tanita Europe).

#### 2.4.2. Dietary Habits and Physical Activity Level

Adherence to a Mediterranean diet was assessed by the Mediet questionnaire, which was analyzed using the DianaWeb Platform and Mediet Score [48]. PAL was assessed by using the interviewer-administered IPAQ-SF questionnaire [49,50]. The questionnaire referred to the last 7 days and asked about walking, moderate-intensity activities, vigorous-intensity activities, and sitting time.

#### 2.4.3. Cardiorespiratory Fitness

Participants’ cardiorespiratory fitness, expressed as maximal oxygen uptake (*V*O_2max_), was estimated using a personalized submaximal incremental walking test performed on a treadmill [32,51]. The submaximal testing protocol consisted of multiple 3 min stages with incremental exercise intensities tailored for each individual according to their predicted *V*O_2max_. The exercise intensity (treadmill speed and grade) of the first stage was set at about 30% of the predicted oxygen uptake (*V*O_2_) reserve (*V*O_2_R) using the ACSM’s walking equation [32] and was increased by about 10% *V*O_2_R each stage. The tests were interrupted when participants reached 70% of heart rate (HR) reserve (HRR) or if safety contraindication or concerns appeared during the test [51]. The HR or *V*O_2_ values corresponding to the desired percentages of the reserve values (%*V*O_2_R or %HRR) were calculated with the following formula: ((maximal value − resting value) x desired percentage) + resting value. In the calculation of %*V*O_2_R, the resting *V*O_2_ was assumed to be 3.5 mL·min^−1^·kg^−1^ [32], while *V*O_2max_ was predicted by means of a non-exercise model using the Excel spreadsheet provided by Ferri Marini et al. [52]. In the calculation of %HRR, the resting HR was measured after a 10 min resting period with the participants sitting quietly in a chair, while maximal HR (HR_max_) was predicted using the formula proposed by Gellish et al. [53]. If the test assumptions and recommendations were met, the *V*O_2max_ of each participant was estimated according to her HR responses during the test by extrapolating her submaximal HR-*V*O_2_ relationship to the predicted HR_max_ [32,51]; otherwise, the test was repeated. The personalized submaximal testing protocols were created at baseline and repeated after the intervention.

#### 2.4.4. Cardiovascular Parameters

Cardiac function was assessed by echocardiography (VIVID 7. GE) and speckle tracking imaging analysis (ECHOPACK version 2.1. GE). Echocardiography was performed and reviewed by a single trained cardiologist. Echocardiographic images were obtained and recorded by standard techniques (parasternal long axis, short axis, apical two chambers, apical three chambers, and apical four chambers) with patients in the left lateral position. Measurements were assessed according to current guidelines [54] and included volumetric measure by the modified Simpson’s rule (left ventricular end-systole and end-diastolic, left atrium end-systolic and ejection fraction), doppler measurement of mitral inflow (E and A wave), tissue doppler lateral mitral annulus peak velocity (e’ wave) and speckle tracking peak global longitudinal strain [54]. HRV was assessed by 24 h monitoring (ELA Medical SyneScope program, version 3.10, Paris, France) [55]. Artifacts or trace suboptimal signals were deleted. Data were excluded for HRV analysis if there was a high burden of premature beats (>5%) or long periods of suboptimal signal (<22 h total valid time). 24 h-holter data included mean heart rate (HR in bpm), total number of premature ventricular and supraventricular beats (as percentage of total beats), time domain HRV parameters (standard deviation of the averaged normal to normal (NN) intervals for all 5 min segments (ASDNN/5 min in ms), root mean square of successive differences of NN intervals (RMSSD in ms), percentage of adjacent NN intervals that varied by more than 50 ms (pNN50 in %) and frequency domain (<0.04 Hz—very low frequency (VLF in ms^2^), 0.05 to 0.15 Hz—low frequency (LF in ms2), 0.15 to 0.4 Hz—high frequency (HF in ms^2^)) and total power (TP in ms^2^) HRV parameters. Circulating cardiac biomarkers such as high sensitivity troponin as reported by Thygersen et al. [56]. An example of the measurements of techniques used to assess the changes in cardiovascular parameters in a representative patient at T0 and T1 is available in Appendix A.

#### 2.4.5. Metabolic, Hormonal, and Inflammatory Parameters

Fasting (≥12 h) blood was obtained from the antecubital vein by trained phlebotomists. Blood glucose, insulin, triglycerides, HDL-C, LDL and total cholesterol concentrations were determined by colorimetric assays on Beckman Coulter AU Analyzers [57,58]. Homeostasis model assessment (HOMA-IR) was used to estimate insulin resistance using the validated equation: Fasting Plasma Insulin × Fasting Plasma Glucose (mmol/L)/22.5 [59]. Progesterone, estradiol and testosterone were determined by chemiluminescence on Beckman Coulter DXi Analyzers [60,61]. Serum high sensitivity C-Reactive Protein (hs-CRP) levels were quantitatively determined by the Beckman Coulter AU System CRP Latex reagent on Beckman Coulter AU Analyzers.

### 2.5. Statistical Analyses

Due to imposed changes in the exercise intervention, hence on the independent variable differentiating IA and CA, preliminary analyses were performed to assess if the two groups showed between group differences over time on several representative clinical functional variables, such as BMI, cardiorespiratory fitness and metabolic prognostic parameters. The IA and CA trend over time showed no statistical differences (Appendix A); hence, the IA and CA participants were combined into a unique group and changes in all samples were assessed using a general linear model (multiple analysis of variance for repeated measures). Comparison between T0 and T1 values of anthropometric, physical fitness variables, body composition, PAL, dietary habits, and cardiac indexes were subsequently compared using univariate tests between subjects cautiously using a conservative Greenhouse–Geisser correction.

Effect sizes were also reported, using Cohen’s d. A commonly used interpretation is to refer to effect sizes as small (d = 0.2), medium (d = 0.5), and large (d = 0.8) based on benchmarks suggested by Cohen [62].

## 3. Results

At T0, participants age and time since diagnosis were 52.6 ± 7.6 years and 10.4 ± 2.9 months, respectively. Baseline participants characteristics are described in Table 1. At 1-year post-intervention, no cardiovascular incidents and no relapses were observed.

### 3.1. Changes in Anthropometric, Body Composition, Physical Activity Level, Dietary Habits, and Cardiorespiratory Fitness

As shown in Table 2, after the 3-month intervention, there were significant ameliorations in BMI, a slight reduction in body weight, whereas waist circumference, and body composition did not change. By contrast the adherence to Mediterranean diet, PAL and *V*O_2max_, and significantly improved.

### 3.2. Changes in Cardiovascular Parameters

At baseline, indicators of an impaired cardiac function, namely LAESV/BSA ≥ 34, GLS > −18%, and e’ ≤ 8 [54], were present in 10 (33.3%), 6 (20.0%), and 6 (20.0%) participants, respectively. Chemotherapy and radiotherapy treatments showed no significant association with the number of participants with impaired baseline GLS (χ^2^ = 0.039, *p* = 0.843 and χ^2^ = 0.103, *p* = 0.748, respectively) and e’ (χ^2^ = 0.039, *p* = 0.843 and χ^2^ = 0.103, *p* = 0.748, respectively), whereas the number of participants showing an impaired LAESV/BSA was higher in patients that underwent chemotherapy (χ^2^ = 4.368, *p* = 0.037) but not radiotherapy treatment (χ^2^ = 3.187, *p* = 0.074).

Ejection fraction was normal in all patients. Subclinical systolic dysfunction (GLS > −18%) was present at T0 in six patients, no significant changes at T1 were observed. Signs of diastolic dysfunction, as defined by guidelines [54] (E/A < 0.8; mitral annulus peak velocity (e’ wave) <9 cm/s or index left atrial volume >34 mL/mq), were present at T0 in 15 patients and significantly decreased (*p* = 0.007; z test for two proportion, paired data), going from 15 to 10 patients on the total sample with at least one sign; no subject in the normal range at T0 showed new signs of diastolic dysfunction after 3 months. During 24 h holter monitoring overall supraventricular and ventricular arrhythmias burden was low in both conditions pre- and post-intervention. A significant reduction in mean heart rate was found after 3 months. Autonomic function assessed by HRV improved in both time and frequency domain as shown in Table 3. No 24 h monitoring was excluded for suboptimal trace or arrhythmias. No association between post-lifestyle intervention and high-sensitivity troponin was found (Table 4).

### 3.3. Changes in Metabolic, Hormonal, and Inflammatory Parameters

Compared with baseline, BCSs experienced significant reductions in most of the metabolic and inflammatory parameters analyzed, including those related to insulin resistance (i.e., glycemia, insulin, and HOMA-IR) and inflammation (hs-CRP). The circulating concentrations of LDL and total cholesterol decreased at T1, whereas the HDL and triglycerides levels did not change compared with baseline. Testosterone levels decreased after intervention, whereas progesterone level did not change. Beta estradiol resulted lower than 20–40 pg/mL in most of the enrolled women, confirming the absence of ovarian function either from cancer therapy or physiologic menopause. Comparison for all biomarkers in response to active and healthy lifestyle as shown in Table 4.

## 4. Discussion

To our knowledge, this is the first study enrolling non-physically active BCSs with high risk of recurrence that analyzes the effects of 3-month home-based lifestyle intervention focused on nutrition and exercise during the Italian pandemic lockdown.

The main result of this study is that the lifestyle intervention significantly improved BMI, cardiorespiratory fitness, metabolic and inflammatory parameters as well as cardiac function indexes and heart rate variability, thus leading to a significant cardiometabolic amelioration. These effects are consistent with the current evidence on lifestyle modification strategies, in particular with proper nutrition habits and exercise training in BCSs, which have been recently described by systematic reviews and meta-analyses [15,63,64,65,66,67,68,69].

In our study, after the intervention, the adherence of the BCSs to the Mediterranean diet and PAL increased by 28.0% and 61.2%, respectively. This supports the feasibility of a 3-month home-based lifestyle intervention during home-confinement.

In the BCSs involved in this study, the lifestyle intervention adherence was associated with a slight improvement in BMI (−1.7%) as well as body weight (−1.2%). As being overweight (≥25 to <30 kg/m^2^) is associated with increased adiposity, particularly visceral adipose tissue, a slight amelioration in body composition could also induce a beneficial effect on cardiometabolic parameters, especially during COVID-19 quarantine, when a worsening in lifestyle routine is expected [37,70,71]. Importantly, the Mediterranean diet proposed in our study was not considered as a weight loss plan, but was beneficial for heart health, decreasing the risk of cardiometabolic disease and potential influence on BC prognosis and prevention of recurrences or secondarisms [72]. Increasing evidence revealed that the protective effects appear to be most attributable to balanced diet, especially with greater adherence to the Mediterranean diet [73].

Secondary/tertiary prevention actions for BCSs should include health lifestyle strategies based on WCRF/ACSM recommendation, including healthy diet (i.e., Mediterranean Diet) and exercise (i.e., with special attention on frequency, intensity, time and type of exercise).

In this regard, participants’ *V*O_2max_ significantly improved after the lifestyle intervention. The average increase found in this study (from 30.5 to 33.4 mL·min^−1^·kg^−1^) is in line with several other studies that show improvements in *V*O_2max_ after on-site [74,75] and remotely [76,77] supervised aerobic exercise trainings, or after combined exercise and nutrition programs [78]. Therefore, even if adapted to cope with the imposed COVID-19 pandemic restrictions, nutrition and exercise interventions seem to be effective in improving cardiorespiratory fitness and thus may represent a useful tool to face with the new health challenges imposed by the COVID-19 lockdown [79] to BCS and BC patients. This assumes particular importance if considering the emerging evidence showing that adjuvant therapy can negatively affect cardiorespiratory fitness [80,81,82] and that cardiorespiratory fitness is usually lower in BC patients compared to healthy controls [83,84].

This study, beyond the physiological effect induced by lifestyle changes, provides interesting indications about cardiometabolic outcomes in a real clinical setting useful for future large-scale initiatives in the field. Regarding cardioprotective effects, at baseline, at least one sign of diastolic dysfunction was present in half of the study population and a systolic subclinical dysfunction (GLS > −18%) was found in 25% of women. These data are related to the previous chemotherapy, radiotherapy and the high prevalence of cardiometabolic risk factors and inflammation due to study selection criteria. Moreover, previous studies reported a persistent worsening of diastolic function with contemporary BC therapy [85]. Diastolic dysfunction is an early marker of higher morbidity and mortality: meta-analytic studies show a 3.5-fold increased risk of cardiovascular events or death in patients with diastolic dysfunction. Precise assessment of diastolic function may be challenging and require multiple invasive and non-invasive testing. Echocardiography is a reliable and widespread tool for such an evaluation. After lifestyle intervention, we found a significant reduction in the number of patients with one or more of the main echocardiography signs of diastolic dysfunction as stated by international guidelines. Autonomic dysfunction is a marker of poor prognosis in different conditions, including cancer patients, cancer survivors [86], and metabolic syndrome, especially in women [87]. HRV through 24 h Holter monitoring is at present the best non-invasive method for assessing autonomic dysfunction, with lower HRV indicating higher dysfunction. HRV is a dynamic biomarker which changes rapidly in response to illness, cardiovascular fitness, mental and physical stress, pain and overall physical wellness [88]. We found a significant improvement in both time and frequency domains, as shown in Table 3. ASDNN/5 min is one of the most reliable parameters of HRV and directly correlated to autonomic function. Total power is the sum of all HRV frequency bands and is linked to total variability and ASDNN. VLF has been correlated with inflammation, with lower levels of VLF indicating higher inflammation [89]. We found a 17% relative increase in VLF at T1 but failed to meet statistical significance. LF is reduced in type II diabetes and improved after the hypoglucidic diet [90]. The significant improvement in LF is in agreement with the amelioration of cardiorespiratory fitness, metabolic profile, and nutrition habits. In addition, cholinergic mechanisms within the inflammatory reflex have, in recent years, been implicated in attenuating obesity-related inflammation and metabolic complications. Recent evidence highlights the relationship among fitness, the autonomic nervous system and metabolic dysfunctions in BC, and their impact on the patient’s prognosis [91].

Our results suggest that a 3-month home-based lifestyle intervention also improves the BCSs’ metabolic condition. Indeed, there were significant reductions in HOMA-IR index (−26.1%) and the glycemia (−9.1%) and insulin (−18.1%) levels. In this line, it is well known that BCSs’ risk of recurrence is increased in women who have attributes of the insulin resistance conditions, such as central obesity, high endogenous insulin levels, clinical metabolic syndrome and physical inactivity. The current evidence linking insulin to BC recurrence is sufficiently compelling so that neoadjuvant and adjuvant intervention studies have been initiated to evaluate clinical anti-cancer effects of exercise and nutrition and actions that modulate insulin levels, as well as other potential non-insulin-mediated anti-cancer effects [63]. Moreover, increasing research has assessed the effect of PA on the HOMA-IR index in BCSs but the evidence is still conflicting [92,93]. Our data agree with the epidemiological study Diet and Androgen-5 study (DIANA-5) [20,39], where a structured 3-month aerobic exercise intervention associated with Mediterranean Diet improves fasting insulin levels, the HOMA-IR index and body composition parameters in BCSs [21]. These data have been confirmed by other recent investigations and systematic reviews [94,95], which confirm the health benefits of PA, healthy eating, and weight management for BCSs.

Overweight and inactivity are also associated with chronic low-grade inflammation and metabolic dysfunction characterized by elevated levels of circulating proinflammatory mediators known to promote tumorigenesis and recurrences [89,90,91,92,93,94,95,96,97,98]. Indeed, metabolic syndrome accompanied by sedentary behaviors associated with being overweight have been established as notable risk factors for BCSs [99,100]. In our study, along with the improvement of cardiometabolic functions, a decreased inflammation was also observed. Indeed, the lifestyle intervention led to a 19.9% reduction in hs-CRP, which is an inflammatory protein positively associated with BC risk recurrence [101,102,103]. BCSs with the state of chronic inflammation are in fact at risk of recurrence and metabolic disturbances even subsequently related to the cardiotoxicity of the chemotherapy [101]. We also noted a reduction in metabolic and hormonal circulating factors such as LDL (−8.4%), total cholesterol (−4.3%) and testosterone (−28.9%), whereas the HDL and triglycerides did not change compared with baseline.

The present study must be analyzed in the context of home-confinement due to the COVID-19 lockdown. The Italian people experienced a quarantine period, with home-confinement that worsened several aspects of the BC cancer care; prolonged staying home has been associated with a sedentary lifestyle, modified diet patterns and higher levels of stress [38,104]. During COVID-19 restrictions, cancer patients and survivors easily regress to sedentary lifestyles, which results in declining health and quality of life, particularly for patients undergoing treatment or suffering adverse effects of treatment [105,106]. In this regard, a recent study showed that 90% of physically active BCSs decreased PAL and increased sedentary time [33]. Moreover, a survey analysis with special attention to understanding the barriers that may influence an active lifestyle in the Italian BC women from the DianaWeb court (781 BC women) underlines the negative impact of the pandemic on MET-min/week of walking, vigorous intensity, and total PA [34]. On the contrary, our results suggest a relevant increase in the total PAL level, despite home-confinement. We found a significant increase in PAL +61.2% after a 3-month home-based lifestyle intervention with an increase of 369 MET-min/week (*p* < 0.001).

Home-confinement made it more difficult to reach the guidelines of oncological prevention for both nutrition and regular PA. Therefore, in this context, home-based lifestyle intervention may represent a valid strategy to better control nutritional habits and mitigate physical inactivity in this fragile population [107].

Importantly, all findings were short-term, and it is unknown if such improvements persist after the 3-month intervention period. Therefore, long-term results are needed to confirm the present study results.

Psycho-sociological distress among cancer survivors involved in this study was out of the attention of this investigation; in further studies, it will be interesting to analyze these aspects in relation with the clinical and functional outcomes.

## 5. Conclusions

The COVID-19 pandemic and associated measures for emergency control had a tremendous impact on several aspects of people’s lives and clinical practices. We observed that a 3-month home-based lifestyle intervention focused on Mediterranean diet and aerobic exercise, which was adapted to the imposed COVID-19 pandemic restrictions, significantly reduced echocardiographic signs of diastolic dysfunction and improved autonomic function. Moreover, there was a significant improvement in BMI, cardiorespiratory fitness, metabolic, and inflammatory parameters, leading to significant cardiometabolic amelioration even during home-confinement. These results seem to confirm the efficacy of multidisciplinary clinical practices that focus on improving BCSs’ lifestyle behaviors as cancer tertiary care even during the COVID-19 lockdown. Future studies, having long-term results on the improvements of the individual health and quality of life of different interventions, are needed to include these interventions in routine clinical practices.

## Figures and Tables

**Figure 1 jcm-10-02678-f001:**
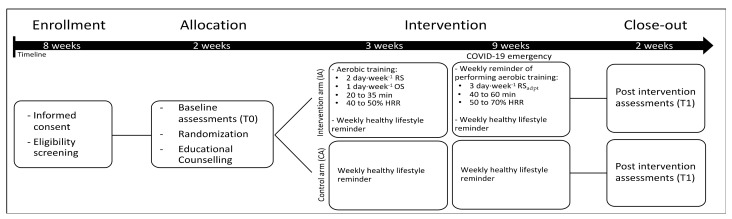
Flowchart of the study design. Abbreviation: HRR, heart rate reserve; RS, remotely supervised; OS, on-site supervised; RS_adpt_, remote supervision adapted due to COVID-19 pandemic restrictions.

**Table 1 jcm-10-02678-t001:** Participants baseline characteristics (*n* = 30).

		*n*	%
Disease Stage			
	0	6	20
	I	15	50
	II	9	30
	III	-	-
Menopausal Status			
	Postmenopausal	18	64.3
Surgery Type			
	Mastectomy	3	10.0
	Quadrantectomy	26	87.7
	Lumpectomy	1	3.3
Treatment in Addition to Surgery			
	Only radiation	2	6.7
	Only chemotherapy	13	43.3
	Radiation and chemotherapy	4	13.3
	None	11	36.7
Current Endocrine Therapy			
	None	6	20.0
	Tamoxifen	8	26.7
	Aromatase inhibitor	16	53.3

Abbreviations: n, number of participants; %, percentage of total number of participants.

**Table 2 jcm-10-02678-t002:** Comparison between pre- (T0) and post-intervention (T1) anthropometric, body composition, physical activity level, dietary habits, and cardiorespiratory fitness parameters.

	T0	T1	Statistics		
	Mean ± SD	Mean ± SD	*p*-Value	ES	%Δ
Weight (kg)	67.1 ± 11.6	66.3 ± 10.9	0.091	0.33	−1.2%
BMI (kg/m^2^)	26.0 ± 5.0	25.5 ± 4.7	0.035	0.41	−1.7%
Waist circumference (cm)	84.4 ± 12.1	84.6 ± 10.7	0.842	0.04	+0.2%
Fat mass (%)	31.1 ± 6.3	30.7 ± 5.9	0.408	0.16	−1.3%
*V*O_2max_ (mL·min^−1^·kg^−1^)	30.5 ± 5.8	33.4 ± 6.8	<0.001	0.91	+9.6%
PAL (MET-min/week)	647 ± 547	1043 ± 564	<0.001	1.03	+61.2%
Adherence to Mediterranean diet (Mediet Score DianaWeB)	6.9 ± 2.3	8.8 ± 2.2	<0.001	1.07	+28.0%

Abbreviations: ES, Cohen’s *d* effect size; %Δ, percentage changes over time; BMI, body mass index; *V*O_2max_, maximal oxygen uptake; PAL, physical activity level.

**Table 3 jcm-10-02678-t003:** Comparison of cardiac function parameters between pre- (T0) and post-intervention (T1).

	T0	T1	Statistics		
	Mean ± SD	Mean ± SD	*p*-Value	ES	%Δ
Echocardiography					
Ejection Fraction (%)	60.8 ± 4.4	62.6 ± 4.1	0.550	0.12	+3.1%
e’ (cm/s)	11.6 ± 3.3	12.0 ± 3.8	0.459	0.15	+3.4%
LVEDV/BSA	47.6 ± 8.7	48.5 ± 8.9	0.015	0.52	+1.8%
LVESV/BSA	18.4 ± 3.5	21.1 ± 11.9	0.119	0.32	+14.6%
LAESV/BSA	31.7 ± 7.8	31.0 ± 9.2	0.350	0.19	−2.0%
GLS (%)	−20.1 ± 2.4	−20.2 ± 2.3	0.569	0.12	+0.6%
24 h-Holter monitoring					
Mean HR (bpm)	76.6 ± 7.8	73.7 ± 8.3	0.003	0.66	−3.8%
Supraventricular extrasystole (%)	0.04 ± 0.1	0.08 ± 0.28	0.380	0.18	+136.9%
Ventricular extrasystole (%)	0.01 ± 0.03	0.02 ± 0.07	0.450	0.15	+115.4%
pNN50 (%)	5.44 ± 5.49	8.03 ± 10.26	0.358	0.19	+47.7%
ASDNN/5 min (ms)	50.6 ± 14.4	55.3 ± 16.7	0.032	0.47	+9.2%
RMSSD	25.12 ± 9.92	26.64 ± 11.65	0.570	0.11	+6.0%
VLF (ms^2^)	1598 ± 967	1881 ± 963	0.118	0.32	+17.7%
LF (ms^2^)	589 ± 391	732 ± 542	0.014	0.53	+24.3%
HF (ms^2^)	157 ± 128	225 ± 241	0.197	0.27	+43.1%
Total power (ms^2^)	2627 ± 1393	3034 ± 1669	0.035	0.42	+15.5%

Abbreviations: ES, Cohen’s d effect size; %Δ, percentage changes over time; e’, lateral mitral annulus peak velocity; HR, heart rate; LVEDV, left ventricle end diastolic volume; LVESV, left ventricle end systolic volume; LAESV, left atrial end systolic volume; GLS, global longitudinal strain; BSA, body surface area; pNN50, proportion of normal to normal (NN) intervals with a difference more than 50 msec; ASDNN/5 min, average of standard deviation of all 5 min NN intervals; RMSSD, root mean square of successive differences of NN intervals; VLF, very low frequency; LF, low frequency; HF, high frequency.3.3. Changes in Metabolic, Hormonal, and Inflammatory Parameters.

**Table 4 jcm-10-02678-t004:** Comparison of metabolic prognostic and inflammatory parameters between pre- (T0) and post-intervention (T1).

	T0	T1	Statistics		
	Mean ± SD	Mean ± SD	*p*-Value	ES	%Δ
Glycemia (mg/dL)	100.8 ± 11.4	91.7 ± 11.0	<0.001	1.15	−9.1%
Insulin (microU/mL)	7.92 ± 4.68	6.49 ± 3.94	0.018	0.47	−18.1%
HOMA Index	2.07 ± 1.54	1.53 ± 1.11	0.005	0.57	−26.1%
Triglycerides (mg/dL)	102.8 ± 43.7	93.3 ± 43.7	0.091	0.32	−9.3%
HDL (mg/dL)	62.3 ± 15.9	60.8 ± 13.5	0.242	0.22	−2.5%
LDL (mg/dL)	137.7 ± 29.9	126.1 ± 28.3	<0.001	0.77	−8.4%
Total cholesterol (mg/dL)	217.7 ± 39.3	208.5 ± 37.3	0.029	0.43	−4.3%
Progesterone (ng/mL)	0.52 ± 0.40	0.50 ± 0.21	0.750	0.06	−4.0%
Testosterone (ng/mL)	0.34 ± 0.27	0.24 ± 0.20	0.003	0.60	−28.9%
hs-CRP (mg/L)	2.18 ± 2.14	1.75 ± 1.74	0.027	0.43	−19.9%
hs-Troponin (ng/L)	3.07 ± 1.14	2.73 ± 2.72	0.479	0.13	−11.0%

Abbreviations: ES, Cohen’s d effect size; %Δ, percentage changes over time; HOMA, homeostasis model assessment; HDL, high-density lipoprotein; LDL, low-density lipoprotein; hs, high sensitive; CRP, c-reactive protein.

## Data Availability

The data presented in this study are available on request from the corresponding author.

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
