# Peer review of "Effects of a Home-Based Lifestyle Intervention Program on Cardiometabolic Health in Breast Cancer Survivors during the COVID-19 Lockdown"

_jcm, 2021, doi:10.3390/jcm10122678_

Round 1
Reviewer 1 Report
More needed in the introduction about breast cancer and the treatment thereof and the effect it has on survivors
will need to improve on the disease parameters description. How many patients had Stage 1 or 2 or 3 tumours. High risk must be stratified by known prognostic factors
Was there any effect noted from radiation on heart function?
How many patients relapsed? was there any clinical effect on cardiovascular incidents?
Author Response
Dear JCM Editor,
please find enclosed our revised manuscript jcm-124625 entitled ‘Effects of a Home-based Lifestyle Intervention Program on Cardiometabolic Health in Breast Cancer Survivors During the Covid-19 Lockdown by Valentina Natalucci, Carlo Ferri Marini, et al. to be considered for publication in Journal of Clinical Medicine.
We carefully took into consideration all reviewers’ comments and we hope that this revised version of the manuscript could be considered suitable for JCM.
All Authors have approved the content and authorship of the revised manuscript.
Here is a point-by-point response to the reviewers’ comments and concerns.
Responses to Reviewers’ Comments
Reviewer 1 - Comments and Suggestions for Authors
Comment 1. ‘More needed in the introduction about breast cancer and the treatment thereof and the effect it has on survivors’
Response 1. In accordance with the reviewer's suggestion, we have added a new paragraph related to breast cancer and the treatment thereof and the effect it has on survivors in the introduction (please, see lines 64-72 and 80-90).
Comment 2. ‘will need to improve on the disease parameters description: How many patients had Stage 1 or 2 or 3 tumours’.
Response 2. We agree with this comment and Table 1 was completed with the required data.
Comment 3. ‘High risk must be stratified by known prognostic factors’.
Response 3. We thank the reviewer for the comment and agree with him, hence we reworded the abstract and lines 136 and 137. We believe that the revised version describes our methods better and is more in line with the citations we used.
Comment 4. ‘Was there any effect noted from radiation on heart function?’
Response 4. According to the reviewer's observation, the significance of the association between chemotherapy (yes / no), radiotherapy (yes / no), and some representative parameters indicating impaired cardiac function was calculated, using chi-square tests. We included this information in the ‘Results’ section (please, see lines 309-316).
Comment 5. How many patients relapsed?
Response 5. So far, at 1-year post intervention, no relapses were observed. This information was included in lines 297-298.
Comment 6. ‘Was there any clinical effect on cardiovascular incidents?’
Response 6. So far, at 1-year post intervention, no effect on cardiovascular incidents was observed. This information was included in lines 297-298.
Reviewer 2 Report
The study is really interesting because takes into account the often-underestimated problem of morbidity and mortality rate of breast cancer survivors.
However, probably due to the imposed changes in the exercise intervention, the study design it is not perfectly clear (what was the activity of the CA once the IA has carried out solely remotely supervised exercise?) Please insert a flow-chart of the protocol carried out by the two groups.
Furthermore, a brief description of the home-based supervised aerobic exercise training and related updated bibliography would be appreciated.
Author Response
Dear JCM Editor,
please find enclosed our revised manuscript jcm-124625 entitled ‘Effects of a Home-based Lifestyle Intervention Program on Cardiometabolic Health in Breast Cancer Survivors During the Covid-19 Lockdown by Valentina Natalucci, Carlo Ferri Marini, et al. to be considered for publication in Journal of Clinical Medicine.
We carefully took into consideration all reviewers’ comments and we hope that this revised version of the manuscript could be considered suitable for JCM.
All Authors have approved the content and authorship of the revised manuscript.
Here is a point-by-point response to the reviewers’ comments and concerns.
Responses to Reviewers’ Comments
Reviewer 2 - Comments and Suggestions for Authors
The study is really interesting because takes into account the often-underestimated problem of morbidity and mortality rate of breast cancer survivors.
Comment 1. However, probably due to the imposed changes in the exercise intervention, the study design it is not perfectly clear (what was the activity of the CA once the IA has carried out solely remotely supervised exercise?) Please insert a flow-chart of the protocol carried out by the two groups.
Response 1. The flow-chart was included to clarify the study design (please, see Figure 1).
Comment 2. Furthermore, a brief description of the home-based supervised aerobic exercise training and related updated bibliography would be appreciated.
Response 2. Additional explanations of the training protocol, along with updated bibliography, have been included (please, see lines 195-197 and 202-209).
Round 2
Reviewer 1 Report
proceed to publication